# A Quasi-Static Quantitative Ultrasound Elastography Algorithm Using Optical Flow

**DOI:** 10.3390/s21093010

**Published:** 2021-04-25

**Authors:** Raphael Lamprecht, Florian Scheible, Marion Semmler, Alexander Sutor

**Affiliations:** 1Institute of Measurement and Sensor Technology, UMIT–Private University for Health Sciences, Medical Informatics and Technology, 6060 Hall in Tirol, Austria; florian.scheible@umit.at (F.S.); alexander.sutor@umit.at (A.S.); 2Division of Phoniatrics and Pediatric Audiology, Department of Otorhinolaryngology, Head- and Neck Surgery, University Hospital Erlangen, Friedrich-Alexander-University Erlangen-Nürnberg, 91054 Erlangen, Germany; marion.semmler@uk-erlangen.de

**Keywords:** elastography, ultrasound, quantitative, phantom study, quasi-static

## Abstract

Ultrasound elastography is a constantly developing imaging technique which is capable of displaying the elastic properties of tissue. The measured characteristics could help to refine physiological tissue models, but also indicate pathological changes. Therefore, elastography data give valuable insights into tissue properties. This paper presents an algorithm that measures the spatially resolved Young’s modulus of inhomogeneous gelatin phantoms using a CINE sequence of a quasi-static compression and a load cell measuring the compressing force. An optical flow algorithm evaluates the resulting images, the stresses and strains are computed, and, conclusively, the Young’s modulus and the Poisson’s ratio are calculated. The whole algorithm and its results are evaluated by a performance descriptor, which determines the subsequent calculation and gives the user a trustability index of the modulus estimation. The algorithm shows a good match between the mechanically measured modulus and the elastography result—more precisely, the relative error of the Young’s modulus estimation with a maximum error 35%. Therefore, this study presents a new algorithm that is capable of measuring the elastic properties of gelatin specimens in a quantitative way using only the image data. Further, the computation is monitored and evaluated by a performance descriptor, which measures the trustability of the results.

## 1. Introduction

Quantitative elastography is a method to evaluate the stiffness of tissue using various imaging techniques. The initial proposal for quantitative ultrasound elastography was published by Ophir et al. [1]. The techniques and algorithms underwent improvements and new methods were developed: the quasi-static approach proposed by Ophir et al. was joined by algorithms such as shear wave elastography, methods using crawling waves and single tracking, among others [2,3]. Additionally, further imaging techniques such as X-ray and computed tomographic elastography were utilized to estimate the elastic properties of tissue. Clinical applications of elastography kept the pace of technical improvements and more and more organs were examined [3] (pp. 7–11).

The basic idea of quasi-static elastography is to apply a small deformation to a sample or the tissue by an ultrasound transducer and record this compression. The estimated displacement can be computed to a strain field in the compressed specimen. This step results in strain images, often synonymously used for elastography [4]. Further, with reasonable stress assumptions, the elastic properties of the evaluated tissue or sample can be computed [5] (p. 62ff). The calculation of quantitative results, e.g., the Young’s modulus, of the tissue needs further assumptions, such as a simplified or a continuum mechanical model [6]. Copious models and applications for different organ systems, such as liver, breast, thyroid, kidney, were developed and used clinically [2]. This algorithm was developed for measuring the elastic properties of vocal folds [7].

Quantitative solutions for the quasi-static elastography can be analytically solved by using the proposals of Sumi et al. [8], Barbone et al. [9], Fehrenbach et al. [10], etc. Iterative methods have been published, such as those by Doyley et al. [11], Oberai et al. [12], Smyl et al. [13], Mohammadi et al. [14], to name just a few. Additionally, deep learning approaches have been proposed [15]. Recent reviews give an overview of the current state of research, such as those by Parker et al. [16], Doyley [6], Sigrist et al. [2], and Alam and Garra [17].

In this work, the compression of homogeneous and non-homogeneous gelatin blocks will be evaluated by a novel quasi-static elastography algorithm. The blocks were compressed with an ultrasound transducer and the compressing force was measured by a load cell. The resulting data were computed with an image registration algorithm using DeepFlow [18]. Further, the strain was calculated by a Savitzky–Golay differentiator [19]. Using this approach, the axial strain component can be computed separately due to the two-dimensional image registration approach. The axial strains are a marker for the degree of bonding between stiffer regions and the surrounding material [4]. This information can, for example, help to distinguish between malignant and benign changes in the tissue [4,20,21]. The lateral strains, perpendicular to the compression direction, are computed as well, enabling the user to gain further information on the lateral tissue movement [22]. The whole registration and strain calculation process is evaluated by a performance descriptor, which automatically chooses the representative frames for further processing [23]. The plane–strain and plane–stress assumptions are only valid when the measurement is performed using very controlled experimental setups, which guarantee well-defined boundary conditions [6]. Finally, the Young’s modulus is calculated by the quasi-elastic approach, using the stress assumption that Love [24] proposed.

The algorithm enables the user to measure the elastic properties of the specimens without access to the radio frequency (RF) signal of the ultrasound device. Therefore, usual CINE frames, which are basically ultrasound video sequences, of the compression process can be used.

The present paper will first introduce the used methods: starting with the governing equations of elastography, the proposed algorithm is explained. Further, the production of the testing specimens and their mechanical measurements as reference for the elastography results is described. Following the Methods section, the results are presented. Conclusively, the results are discussed and compared to other algorithms.

## 2. Methods

The elastic modulus of the specimens was measured mechanically and by the elastography algorithm, while the mechanical measurements were used as reference values. This section is structured as follows: first, a brief introduction to the theory of linear materials will be given. Secondly, the measurement setup will be presented. Further, the production of the gelatin specimens and, conclusively, the measurement methods and evaluation algorithms will be explained. Two types of mechanical measurements, compression and indentation, were carried out to estimate the elastic modulus of the specimens. The results were analyzed with a finite element (FE) model. At the end of this section, the elastography algorithm will be presented.

### 2.1. Linear Continuous Materials

We will use different models to describe the behavior of the gelatin block, but all models are based on the same assumptions. We assume that gelatin is an isotropic, nearly incompressible, linear–elastic, and locally homogeneous material [2]. The contact surfaces are considered to be a non-slip boundary [25]. The linear formulation of Hooke’s Law with strains ε and stresses σ is
(1)σ=C·ε,
with the 4th-order stiffness tensor *C*, which describes the material’s reaction to stress [26,27]. For orthotropic, (transversal) isotropic materials, *C* simplifies drastically. All elements of *C* except
(2a)$$C_{xxxx}=C_{yyyy}=C_{zzzz},$$
(2b)$$C_{yzyz}=C_{xzxz}=C_{xyxy},$$
(2c)$$C_{xxyy}=C_{xxzz}=C_{yyzz}=C_{yyxx}=C_{zzxx}=C_{zzyy}$$
are zero. Therefore, the stiffness tensor is defined by two values, namely the Poisson’s ratio ν and the shear modulus *G*. The nonzero entries of *C* are consequently defined by
(3a)$$C_{xxxx}=\frac{2(1-\nu )G}{1-2\nu },$$
(3b)$$C_{yzyz}=G$$
and
(3c)$$C_{xxyy}=\frac{2\nu G}{1-2\nu }.$$

The Poisson’s ratio ν is calculated through the ratio of compressing strain to the expansion in the perpendicular direction, and hence it is defined by
(4)$$\nu =-\frac{\varepsilon_{zz}}{\varepsilon_{xx}}.$$

The Young’s modulus is defined as follows:(5)E=2G(1+ν),
and therefore a Hookean material is fully described by the Young’s modulus *E* and the Poisson’s ratio ν [26].

The material can be described as linear–elastic, when only considering small deformations [6]. It is further assumed that the stress has no perpendicular components (*x* and *y*-direction) to the axial compressing stress in the z-direction. The underlying idea is that the tissue expands in perpendicular directions, preserving the volume of the tissue [5]. Therefore, we can define the axial stress $\sigma_{0}$ with
(6)$$\sigma_{zz}:=\sigma_{0}\neq 0,$$
when all other components of the stress are zero ($\sigma_{ij}=0$). Hooke’s Law (Equation (Equation 1)) can be written, using the axial strain $\varepsilon_{0}$, as
(7)$$\varepsilon_{zz}=\frac{1}{E}\sigma_{0}:=\varepsilon_{0}.$$

Conclusively, we can compute the elastic modulus *E* with
(8)$$E=\frac{\sigma_{0}}{\varepsilon_{0}}.$$

In non-linear materials, the measured strain depends on the applied stress or deformation. Therefore, a strain-dependent elastic modulus can be modeled by a Veronda–Westmann material [5]. In the special case of small strains and uniaxial stress, their relation can be described by
(9)$$\sigma_{0}\approx E\varepsilon_{0}e^{3\gamma \varepsilon_{0}^{2}}\;\;\mathrm{for}\;\varepsilon_{0}\ll 1.$$

The non-linearity parameter γ of the apparent Young’s modulus is a tissue-dependent material parameter [5]. Using this, we can define a strain-dependent Young’s modulus
(10)$$E_{NL}=Ee^{3\gamma \varepsilon_{0}^{2}}.$$

The values chosen in this paper are γ=10 and ν=0.495, which have yielded good results [28,29].

### 2.2. Setup

The measurement setup consists of three basic components, which are displayed in Figure 1: a linear drive powered by a stepper motor, a load cell (KD23s-2N, ME-Messsysteme, Hennigsdorf, Germany) and the ultrasound transducer (IO 8-17, Alpinion, Anyang, Korea) connected to the ultrasound device (E-Cube 15EX, Alpinion, Anyang, Korea). The ultrasound transducer has a frequency range of 8 MHz to 17 MHz.

The CINE frames were recorded at a central ultrasound frequency of 15 MHz with a lateral and axial pixel size of 0.045
mm. The frame rate was 43±3 frames/s, whereas the exact frame rate depends on the ultrasound device and cannot be influenced directly.

The specimen was placed on a solid baseplate and compressed by the transducer (elastography and indentation measurements) or a stiff plate (compression measurement). The baseplate was coated with sandpaper (320 grid) to ensure the non-slip boundaries. In the case of the compression measurements, the top plate was coated in the same way. The whole measurement process was controlled and monitored by Matlab (2020a, MathWorks, Natick, MA USA). The stepper motor was driven by a micro controller (Tic500, Pololu, Las Vegas, NV, USA) with a serial interface, which also stored the covered steps. During the compression, the load cell measured the force and the data were recorded by an analog input module (NI-9219, National Instruments, Austin, TX, USA). Simultaneously, the start of the CINE sequence was triggered.

### 2.3. Specimen Preparation

Extensive research has been done on tissue-mimicking phantoms for ultrasound elastography using different materials [30]. Water-based phantoms are made of gelatin and/or agar [31,32] as natural matrix materials, whereas also synthetic polymers have been used [30]. To achieve better imaging quality and contrast, solid scatter particles are added. Their size must be chosen carefully according to the ultrasound frequency. Mostly, particles with a diameter of 0.5μm–50μm are used [30,31,32]. For this purpose, glass beads [31], silica [32], talc powder [33] etc., have been successfully tested. Further additives can be used to avoid bacterial colonization or to increase the cross-link between the gelatin molecules [31].

The gelatin blocks were cast in a mold, which measured 25 mm × 25 mm × 45 mm; see Figure 2. One softer and one harder mixture was produced (Table 1). The gelatin concentration can be assumed to be proportional to the square root of the Young’s modulus of the resulting specimen [11]. Three different kinds of specimens were produced: soft, hard and with inclusion, which consisted of both mixtures. The non-homogeneous specimens had a cylindrical inclusion with a diameter of 8 mm. The soft samples are called $S_{1\ldots n}$, the stiffer ones $H_{1\ldots n}$ and samples with inclusion $I_{1\ldots n}$. Furthermore, the different measurements were numbered chronologically M${}_{1\ldots n}$.

The production process was the same for all samples, which followed Yengul et al. [33]. First, gelatin from porcine skin (SKU G2500 300 Bloom Type A, Sigma-Aldrich, Vienna, Austria) was hydrated with distilled water and heated up to 45 ${}^{\circ }C$ in a water bath. After the clarification of the solution, the glass beads (SiLibeads SOLID 0-50 my, Sigmund Lindner GmbH, Warmensteinach, Germany) were added. The solution was stirred with a magnetic stirrer for an additional five minutes. Thereafter, the gelatin was cast in the prepared mold, which was lubed with petroleum jelly (TRVAS90, Silikonfabrik, Ahrensburg, Germany). The casts were rotated by a stepper motor with 5 rotations/min, in order to avoid sinking of the glass beads.

The specimens were stored at 4 ${}^{\circ }C$ overnight. After 24 h in the fridge, the placeholder for the inclusion was removed. Next, the harder mixture was prepared according to the explained recipe. It was cast into the space for the inclusion and in an additional mold for a homogeneous specimen. Before measurement, the specimens were stored for 24 h in the fridge. The storage time was chosen to be shorter than in Yengul et al. [33], due to the swelling of stiffer inclusions over time [34].

The measurement protocol for all specimens was identical. After resting in the fridge, the specimens were stored at room temperature ( 23 ${}^{\circ }C$) for 5 h. The first measurements were five iterations of the compression measurement. After this, five iterations of elastography measurements were performed, which were followed by the indentation measurements. Therefore, for every sample, 15 measurements ($M_{1-15}$) were carried out. The measurement duration for the stepwise indentation and compression measurements was around 10 min each and the elastography measurement took around 2 min.

### 2.4. Mechanical Reference Measurements

The mechanical measurements were carried out to obtain reference data for the elastography results. In this section, the different evaluation methods for the mechanical measurements are presented—first, the compression measurement, followed by the indentation measurements.

#### 2.4.1. Compression Test

The compression measurements were carried out with a stiff plate larger than the top surface of the specimen. These measurements are used to evaluate the compression modulus—Equation (Equation 11)—of the homogeneous gelatin blocks. The specimen was compressed step-wise, in which every step took 1 s, with a step width of 0.2
mm. The displacement was estimated by the steps of the stepper motor driving the linear drive and the compression force *F* was measured after every single step. Around 20 data points were recorded; therefore, the overall compression process per iteration took around 20 s per iteration. Exemplary results can be found in Figure 3.

The Young’s modulus *E* can be determined by the relation for the compressing stress
(11)$$E=\frac{3\sigma }{(\lambda^{-2}-\lambda )Z}.$$

Here, *Z* is a shape factor, which is related to the ratio *S* of one bonded to the free surface, by $Z=(1+2S^{2})$. Furthermore, λ is the ratio of strained to unstrained height of the specimen [35] (p. 155f). The samples were compressed up to a strain rate of 15%. Above this strain rate, the expanding part of the sidewalls could touch the compressing plate and thus the compressing area enlarged [36]. Although the strain rate is close to that value, in our measurement, such behavior was not visible.

#### 2.4.2. Indentation Test

For the indentation measurements, the homogeneous specimen was compressed step-wise by the ultrasound transducer. The data acquisition was done in the same way as in the compression measurements. The Bulychev–Alekhin–Shorshorov (BASh) relation—Equation (Equation 12)—was used to evaluate the indentation modulus of the specimens [26] (p. 33).

The experimental estimation of *E* was done by an indentation test with a rectangular indenter, namely the ultrasound transducer. Therefore, the generalized BASh relation
(12)$$\frac{dF}{d\delta }=\phi_{c}\frac{2}{\sqrt{\pi }}\sqrt{A}M_{3}^{\prime }$$
was used to estimate the indentation modulus $M_{3}^{\prime }$, in which *F* is the compression force, δ the displacement of the indenter, $\phi_{c}$ the contact area shape factor and *A* the contact area [26] (p. 33).

The contact area shape factor $\phi_{c}$ defines the difference from a circular indenter and can be derived analytically [26] (p. 34). The value for rectangular intenders varies between $\phi_{c}\;\approx \;1.012$ [37] and 1.016 [38]. The ultrasound transducer is not an exact rectangle and it deforms itself. Nevertheless, a value of $\phi_{c}=1.016$ fits the results estimated by a finite element simulation.

Conclusively, the Young’s modulus *E* can be calculated with
(13)$$E=M_{3}^{\prime }\left(1-\nu^{2}\right).$$

#### 2.4.3. Finite Element Model

In this section, we will examine finite element modeling of homogeneous gelatin blocks under compression. The blocks were compressed either by the ultrasound transducer or by a large compressor. The finite element models (Figure 3) of the samples and both compression processes were designed with the GIBBON toolbox [39] (v3.5) for Matlab and the FE model was solved by FEBio [40] (v3.0.1) and its optimization module. This method is understood as the most exact, due to the complex boundary conditions, which are not considered to full extent in the simpler models described before [25,36].

The FEBio material type *Neo-Hookean* was chosen, which describes non-linear stress–strain behavior with two material constants—the Young’s modulus *E* and the Poisson’s ratio ν. In the case of small strains, as in ours, this model reduces to the classical linear elasticity [41]. The governing equation, describing the hyperelastic strain energy function *W*, is
(14)$$W=\frac{\mu }{2}\left(I_{1}-3\right)-\mu \ln J+\frac{\lambda }{2}{\left(\ln J\right)}^{2}.$$
where λ and μ are the first and second Lamé parameters, respectively. *J* indicates the determinant and $I_{1}$ the first invariant of the right Cauchy–Green deformation tensor [40,41].

The defined boundaries of the model describing the compression with the ultrasound transducer are shown in Figure 3. Regarding the compression test, the whole top surface was chosen as the displacement boundary. The measured displacement defines the movement of the boundary nodes shown in green in Figure 3. The nodes indicated in red are the fixed bottom of the model. The model is well defined for a forward simulation with these boundaries (Figure 4). To estimate the elastic modulus of the gelatin block, an inverse approach was applied. The built-in optimization module of FEBio was used, which minimizes the function
(15)$$f\left(a\right)=\sum_{i=1}^{n}{\left[y_{i}-y\left(x_{i};a\right)\right]}^{2}.$$

The optimization module seeks the minimum of f(a), in which the data points $\left(x_{i},y_{i}\right)$ are user-defined and $y(x_{i};a)$ are calculated by the forward model [40]. The Levenberg–Marquardt method is utilized to find this minimum [42,43]. In our case, the reaction force at the transducer gelatin interface is used as the optimization input and the elastic modulus of the gelatin is the optimization target a.

### 2.5. Elastography

The proposed elastography workflow to calculate the Young’s modulus of the specimen is displayed in Figure 5. The specimens were compressed by the ultrasound transducer up to a compression depth of 3 mm measured from the top surface. The compression should last around one quarter up to one second [5] (p. 62ff). Due to the limited frame rate of the ultrasound machine, the compressing speed was set to 1 mm/s.

The evaluation algorithm consists of four basic steps: the displacement (1) between the single frames needs to be estimated; the strain (2) is computed, which is combined with a reasonable stress assumption (3); to conclusively calculate the Young’s modulus (4) with Equation (Equation 8). These steps will be presented in the following sections.

#### 2.5.1. Displacement Estimation

Most elastography algorithms work with the RF signal and are able to detect movements up to a few micrometers. In our case, we use an image registration algorithm to detect the movements [44]. The intrinsic assumption is that the acoustic properties of the probe are constant over the whole measurement process [5,45].

The optical flow algorithm DeepFlow [18] was used to estimate the displacement u between the ultrasound frames. The images were preprocessed using a Gaussian Blur, because this is assumed by the DeepFlow algorithm [18]. The displacement field describes the movement of every tissue part between the unloaded and loaded state. Working with image data and a dense optical flow algorithm, the tissue is discretized by the pixels of the ultrasound frames. Since the displacement between the unloaded frame and every flowing frame needs to be calculated, large displacements could occur. To avoid the failure of the algorithm, the performance after every frame is evaluated by correlation of the reference frame and the warped target frame, using the estimated displacement. The correlation coefficient is calculated by
(16)$$\rho =\frac{\sum_{m}\sum_{n}\left(A_{mn}-\bar{A}\right)\left(B_{mn}-\bar{B}\right)}{\sqrt{\left(\sum_{m}\sum_{n}{\left(A_{mn}-\bar{A}\right)}^{2}\right)\left(\sum_{m}\sum_{n}{\left(B_{mn}-\bar{B}\right)}^{2}\right)}}$$
in which *A* and *B* are the ultrasound frames k−1 and *k*, respectively, and $\bar{A}$ is the mean of *A* [46]. The $k^{\mathrm{th}}$ frame is warped with the estimated displacement field u; hence, ρ is an indicator of the quality of the optical flow field. If ρ falls below a predefined value, the current frame is used as the new reference frame and the displacement calculated so far is added to all further displacement fields; see Figure 6 [47]. The minimum correlation coefficient is chosen $\rho_{min}=0.9$.

#### 2.5.2. Strain Estimation

The strain in the tissue can be calculated by deriving the displacement field
(17)$$\nabla u\left(x,t\right)=\left(\begin{array}{cc} \frac{\delta u_{x}}{\delta x} & \frac{\delta u_{x}}{\delta z}\\ \frac{\delta u_{z}}{\delta x} & \frac{\delta u_{z}}{\delta z} \end{array}\right)$$
where $u_{x}$ and $u_{z}$ are the *x*- and *z*-components of the displacement field [5]. Therefore, two consecutive displacement fields are computed, and this process is illustrated in Figure 7.

The displacement field is noisy, due to random changes in the image data. Moreover, the derivation during strain computation magnifies the primary error. Thus, the displacement field needs to be smoothed or filtered [48,49,50]. The strain can be directly derived, with the mentioned shortcomings, or filters can be used, such as the Least-Squares strain estimator [51]. We use a Savitzky–Golay differentiator to compute the strain, which smooths the displacement field additionally [19]. The 2D Savitzky–Golay differentiator is given by
(18)$$h\left(x,z\right)=\frac{3}{{\left(2M+1\right)}^{2}\left(M+1\right)M\times GS}\times \left[\begin{array}{ccccccccc} -M & -(M-1) & \cdots & -1 & 0 & 1 & \cdots & M-1 & M\\ \vdots & \vdots & \; & \vdots & \; & \vdots & \; & \vdots & \vdots \\ -M & -(M-1) & \cdots & -1 & 0 & 1 & \cdots & M-1 & M\\ \vdots & \vdots & \; & \vdots & \; & \vdots & \; & \vdots & \vdots \\ -M & -(M-1) & \cdots & -1 & 0 & 1 & \cdots & M-1 & M \end{array}\right].$$

The applied filter width *M* determines the degree of smoothing and GS indicates the grid step, which is defined by the pixel width of the ultrasound images. Further, the strain $\varepsilon_{xx}$ at the point (i,j) can be calculated by
(19)$$\varepsilon_{xx}\left(i,j\right)=\sum_{x=-M}^{M}\sum_{z=-M}^{M}h\left(x,z\right)u\left(i+x,j+z\right).$$

The estimated strain is used to calculate the Young’s modulus and the Poisson’s ratio, where the latter is computed by Equation (Equation 4).

#### 2.5.3. Stress Distribution

There is no measurement method known that is able to quantify the internal stress distribution. Therefore, the stress distribution is usually assumed constant [6]. Love proposed an analytic solution to calculate the stress distribution in a semi-infinite, isotropic and homogeneous specimen loaded with a rectangular compressor [24,52,53].

Here, we present only the solution, as the complete derivation is cumbersome and can be found in the work of Love [24]. The stress $\sigma_{zz}$ in compression direction *z* can be written as
(20)$$\begin{array}{c} \sigma_{zz}=\frac{1}{2\pi }\left(\frac{\partial V}{\partial z}-z\;\cdot \;\frac{\partial^{2}V}{\partial z^{2}}\right) \end{array}$$
where *V* is the Newtonian potential. The derivatives can be explicitly solved, depending on the compressing pressure *p* and the geometric dimensions of the compressor 2a and 2b; see Figure 8. The first partial derivative $\frac{\partial V}{\partial z}$ can be written as
(21)$$\begin{array}{c} \frac{\partial V}{\partial z}=-p\;\cdot \;\Omega , \end{array}$$
with
(22)$$\begin{array}{c} \Omega =2\pi -\cos^{-1}\frac{\left(a-x)(b-y\right)}{\sqrt{{\left(a-x\right)}^{2}+z^{2}}\sqrt{{\left(b-y\right)}^{2}+z^{2}}}-\cos^{-1}\frac{\left(a-x)(b+y\right)}{\sqrt{{\left(a-x\right)}^{2}+z^{2}}\sqrt{{\left(b+y\right)}^{2}+z^{2}}}-\\ \cos^{-1}\frac{\left(a+x)(b-y\right)}{\sqrt{{\left(a+x\right)}^{2}+z^{2}}\sqrt{{\left(b-y\right)}^{2}+z^{2}}}-\cos^{-1}\frac{\left(a+x)(b+y\right)}{\sqrt{{\left(a+x\right)}^{2}+z^{2}}\sqrt{{\left(b+y\right)}^{2}+z^{2}}}. \end{array}$$

In this, *x*, *y* and *z* are the coordinates of the current observation point. The second partial derivative of *V* results in
(23)$$\begin{array}{c} \frac{\partial^{2}V}{\partial z^{2}}=p\;\cdot \;[\frac{a-x}{{\left(a-x\right)}^{2}+z^{2}}\left(\frac{b-y}{a_{1}}+\frac{b+y}{d_{4}}\right)+\frac{a+x}{{\left(a+x\right)}^{2}+z^{2}}\left(\frac{b-y}{b_{2}}+\frac{b+y}{c_{3}}\right)+\\ \frac{b-y}{{\left(b-y\right)}^{2}+z^{2}}\left(\frac{a-x}{a_{1}}+\frac{a+x}{b_{2}}\right)+\frac{b+y}{{\left(b+y\right)}^{2}+z^{2}}\left(\frac{a-x}{d_{4}}+\frac{a+x}{c_{3}}\right)], \end{array}$$
with $a_{1}$, $b_{2}$, $c_{3}$ and $d_{4}$ indicating the distance of the observation point to the corners *A*, *B*, *C* and *D*, respectively. The pressure *p* is calculated by dividing the applied force *F* by the compressed area *A*.

We consider two compressors at the top (transducer) and at the bottom (floor) of the phantoms. Due to this, the effect of both can be superposed with
(24)$$\sigma_{total}=\sigma_{top}+\frac{A_{1}}{A_{2}}\sigma_{bottom}$$
where $A_{1}$ is the surface of the compressing transducer and $A_{2}$ the base area of the specimen [54,55,56].

In Figure 9, Love’s solution (Figure 9a) is compared to a finite element simulation of a homogeneous sample (Figure 9b). It can be seen that the stress field is quite similar. However, the relative error is still high (Figure 9c), but lower compared to the plane-stress assumption (Figure 9d).

The stress estimation with Love’s solution is valid for homogeneous elastic materials. Nevertheless, tissue and phantoms could and will have inclusions or stiffer parts. Therefore, we will compare the prediction of Love’s solution with the finite element simulation of a specimen with a stiffer inclusion in Figure 10. The stress estimations with Love and the FE model are shown in Panel (a) and (b), respectively. The relative error between the FEBio result and the stress estimations is indicated in Panels (c) and (d). The difference between the two assumptions shows that, especially in the inclusion’s case, Love’s solution fits better.

Figure 11 shows the mean Young’s modulus of a homogeneous specimen calculated by the presented algorithm, in which Figure 11a shows the results of the plane stress assumption and Figure 11b the one of Love’s assumption. It can be clearly seen that the estimated elastic modulus is much more homogeneous with the assumptions of Love.

#### 2.5.4. Performance Descriptor

Different approaches to dynamically choose the frames, which are taken into account for further processing, were proposed [4,23,57]. To evaluate the reliability of the strain field and the elastography data, a performance descriptor, inspired by Jiang et al. [23], was used. Jiang et al. proposed to measure the strain performance $p_{S}$ with
(25)$$p_{S}=\rho_{RF}\rho_{s},$$
where $\rho_{RF,\;s}$ are the normalized correlation coefficients—see Equation (Equation 16)—between the ${\left(k-1\right)}^{\mathrm{th}}$ and $k^{\mathrm{th}}$ frame of the RF and the strain data, respectively. In order to achieve a meaningful correlation, the strain and RF data of the $k^{\mathrm{th}}$ frame were warped using the estimated displacement field. The performance indicator $p_{S}$ varies between 0 and 1, whereat a minimum value of $p_{min}=0.9$ was considered as trustworthy. In our case, $\rho_{RF}$ was replaced with $\rho_{i}$, using the image data, which represent the computed RF data. After the strain estimation, the reliability of the results was evaluated with Equation (Equation 25). The strain frames which could not fulfill the minimum performance requirement $p_{S}<p_{min}$ were discarded, as illustrated in Figure 12.

Extending the descriptor of Jiang et al. [23], we used a similar descriptor for the elastography algorithm. The reliability of the elastography data was computed in the same way as $\rho_{s}$ for the strain data, and the resulting descriptor was called $\rho_{e}$. Therefore, the overall performance $p_{E}$ is given by
(26)$$p_{E}=\rho_{i}\rho_{s}\rho_{e}.$$

#### 2.5.5. Poisson’s Ratio and Young’s Modulus

The presented algorithm calculates the strain in all four directions. Therefore, using the axial and lateral components, the spatial distribution of the Poisson’s ratio can be computed [58]. It is calculated by Equation (Equation 4), whereby Fehrenbach [10] showed that the two-dimensional Poisson’s ratio, now indicated as $\nu^{\prime }$, is related to the three-dimensional Poisson’s ratio by
(27)$$\nu =\frac{\nu^{\prime }}{1-\nu^{\prime }}.$$

At the end, the Young’s modulus *E* of the sample can be calculated. This is done by using Equation (Equation 8) in a framewise manner. The strain $\varepsilon_{0}$ is estimated by the Savitzky–Golay Differentiator Equation (Equation 19), whereas only frames which fulfill the performance requirements of Equation (Equation 25) are considered. The stress $\sigma_{0}$ is computed by Equation (Equation 24). The force *F* was continuously measured by the load cell and the compressing areas $A_{1,2}$ were measured beforehand.

The results of the elastography algorithm were evaluated by the overall performance $p_{E}$ using Equation (Equation 26). Conclusively, the mean Young’s modulus over all frames was calculated. Therefore, the framewise results are warped on the original configuration using the estimated displacement field. In this process, all frames which did not meet the minimum performance were discarded.

#### 2.5.6. Data Visualization

The results of the elastography were compared to the mechanical results. Hence, regions of interest were chosen and the mean value of the estimated Young’s modulus in these regions was calculated [4]. The number of regions depends on the nature of the specimen: for homogeneous samples, one region and for specimens with inclusion three regions were defined. The selected regions of interest are shown in Figure 13.

## 3. Results

### 3.1. Mechanical Measurements

The Young’s modulus *E* was estimated by the three algorithms explained in Section 2.4: indentation test, compression module and finite element model. The corresponding results are shown in Figure 14 and Table 2. The samples are named according to Table 1.

### 3.2. Elastography Measurements

The elastography results of the specimens with inclusion will be presented in the following section. Firstly, the displacement for the most compressed frame of I${}_{4}$ is indicated in Figure 15.

Secondly, the measured strain in the specimen is displayed for the same frame in Figure 16. The strain $\varepsilon_{zz}$ (Figure 16d) in the compressing direction is the dominant component, which is further used for the Young’s modulus computation.

Using the strain components $\varepsilon_{zz}$ and $\varepsilon_{xx}$, the Poisson’s ratio is calculated using Equation (Equation 4) and (Equation 27); these results are shown in Figure 17.

The performance of the algorithm is shown in Figure 18, in which the minimum correlation of $p_{min}=0.9$ is indicated.

In Figure 19, the mean Young’s modulus of a specimen with an inclusion of 8 mm diameter is shown. Therefore, the expected modulus estimated by the mechanical reference measurements (Table 2) is displayed in the upper left panel (Figure 19a). Further, five measurement iterations are depicted in the following panels (Figure 19b–f).

In clinical applications of elastography, it is more common to use the shear wave velocity $c_{t}$ as an indicator of the tissue properties. Although this is a quasi-static approach, the units can be converted under the assumption of the material model used. Therefore, the shear wave velocity is calculated by $c_{t}=\sqrt{MME\;/\;2\rho \;\cdot \;(1+\nu )}$, in which the density is assumed to be ρ = 1000 kg/$m^{3}$ [59]. The resulting velocity is indicated in Figure 20.

The elastography results were further evaluated in the regions of interest, which are shown in Figure 13. The mean value of every iteration is indicated as a data point in Figure 21. The median value over all iterations is displayed as a red line. The exact values for all samples or regions are stated in Table 3.

The expected and measured Young’s modulus are directly compared in Figure 22 at the two observation depths shown in Figure 22a. The five consecutive iterations $M_{6-11}$ are displayed in Figure 22b.

## 4. Discussion

In this paper, we present an elastography algorithm that can be easily implemented without accessing the raw RF data of the imaging device. This makes the algorithm more widely applicable because it is not limited to research systems. The use of a CINE sequence in any compression process enables the user to display strain images, often vaguely referred to as (strain) elastography. The additional force measurement allowed quantitative results. Quantitative algorithms for ultrasound elastography have been presented before, starting from the first proposals by Ophir et al. [1] to recent works [60]. The quantitative algorithms require either a stress indicator or a force measurement.

The underlying material model relies on assumptions, which possibly do not reflect the real material. The Young’s modulus is not constant over strain rates of 15%; in our case, we do not exceed this limit so linear approaches are sufficient [61]. The linearity of the examined gelatin samples was experimentally confirmed by the mechanical measurements, where the samples showed linear behavior up to a 15% strain rate. Nevertheless, we take non-linear behavior into account with an approximation of a non-linear material model; see Equation (Equation 10). The non-linearity parameter was chosen to be γ=10, as reported in the associated literature [5]. The method of Goenezen et al. [28] could be applied to estimate γ for gelatin and other materials or tissue. Further experiments could also validate the assumption of near incompressibility, by measuring the Poisson’s ratio, although, as Fehrenbach [10] pointed out, the exact value of the Poisson’s ratio has little influence on the resulting Young’s modulus. The isotropy and local homogeneity are properties which were not further evaluated in this study.

The elastography results show good agreement with the mechanically estimated Young’s modulus. Comparing the median values from Table 2 and Table 3 for the non-homogeneous specimen, the relative error for the inclusion and the body is 34% and 24%, respectively. Taking a closer look at Figure 21, the Young’s modulus of the inclusion seems to be drastically underestimated. This is in contrast to the expected hardening of the inclusion over time, due to the diffusion of water. Further, it would be expected that the inclusion would appear harder because of target hardening artifacts [62]. Therefore, the possible reason for this artifact could be the temporal course of the measurement, in which the elastography measurement was always the last one. The drying of the specimen increases its hardness, whereas the inclusion was protected from this process.

Although elastic behavior is a reasonable assumption for quasi-static processes, the relatively fast compression of 1 mm/s during the elastography measurement could push the assumption to its limits. The viscoelastic properties, which imply that the strain and stress are out-of-phase, may become relevant. In particular, in the first few seconds after the compression, the force and therefore the stress decrease rapidly, which can be traced back to a creep in the material [63]. During the mechanical reference measurements, the specimen had more time to find a stable state, whereas during the elastography measurement, the force was measured immediately. Further, the inertia of the specimen might play a role [64]. Therefore, the viscoelastic nature of gelatin and tissue will be taken into account for further algorithms.

The stress indicator is determined to obtain a reference stress value or Young’s modulus to transform the strain ratios into quantitative results [1,8]. In this proposal, an additional load cell was used, which could be replaced by a stress indicator, e.g., an elastic reference layer between the tissue and the transducer. This would make the algorithm more widely applicable.

The extensive assumptions of Love’s stress estimation lead to shortcomings in the approach. First, the finite specimen contradicts the assumption of an infinite elastic domain. This assumption is valid for specimens which are at least four times larger than the compressor [65]. Second, the material is considered relatively homogeneous; this assumption is valid for modulus contrasts of ±6 [54]. For larger contrasts, the stress distribution is not properly described by Love’s solution. A solution could be contrast-transfer efficiency, proposed by Ponnekanti et al. [66]. Third, we assume that the compressor is uniformly loaded, whereas in reality, the surface is uniformly displaced, which corresponds to nonuniform loading [54,67]. The largest error due to this is caused close to the compressed surface [24,54,65].

In Figure 9 and Figure 10, the results for $\sigma_{zz}$ with Love’s approach are shown. For comparison, the results of FEBio for the same boundary conditions are indicated in the same figures. Some of the assumptions in Love’s approach are not met, e.g., the dimensions of the specimen are not four times the transducer size. Due to this, the match of the estimated stress fields is not perfect and the finite element approach may map the real stress distribution better.

The stress assumptions avoid the limitation of uniform stress, the so-called target hardening, by computing non-uniform stress with the stress solution of Love [24,56,65]. Therefore, the stress is realistically adopted and maps the Young’s modulus with high precision. The modulus contrast could have also been calculated by the function for the modulus contrast for cylindrical inclusions, which Kallel et al. [68] proposed and was widely used [34,54,60]. In order to make the algorithm more universal and applicable to non-cylindrical inclusions, we waived this approach. Unfortunately, the stress assumption that we used does not take into account the fact that the specimen has a varying Young’s modulus. Hence, artifacts such as shadowing artifacts occur and are visible in Figure 19 [69]. Additionally, the stiffer specimen could itself shadow the ultrasound image. Using our mixture, this artifact was not visible.

Various algorithms for a performance evaluation of strain data were proposed [4,23,57,70]. The presented descriptor is quick to compute and can be easily extended to the description of the elastography data performance. Hence, it is an easy-to-use indicator for the user and for further computation. The indicator and therefore the trustworthiness of the result are highlighted for the user.

The presented algorithm was able to detect the inclusion and measure its Young’s modulus with a precision of a few Kilopascal. The whole algorithm is based on very few data; therefore, comprehensive assumptions were made. Nevertheless, the results support these assumptions. Using this algorithm in vivo, some adaptions may be necessary, due to the changed boundary conditions in the tissue, e.g., a non-axial compression would alter the results and violate the assumptions made. The proposed algorithm will be evaluated on tissue in the near future to test its applicability and accuracy.

## Figures and Tables

**Figure 1 sensors-21-03010-f001:**
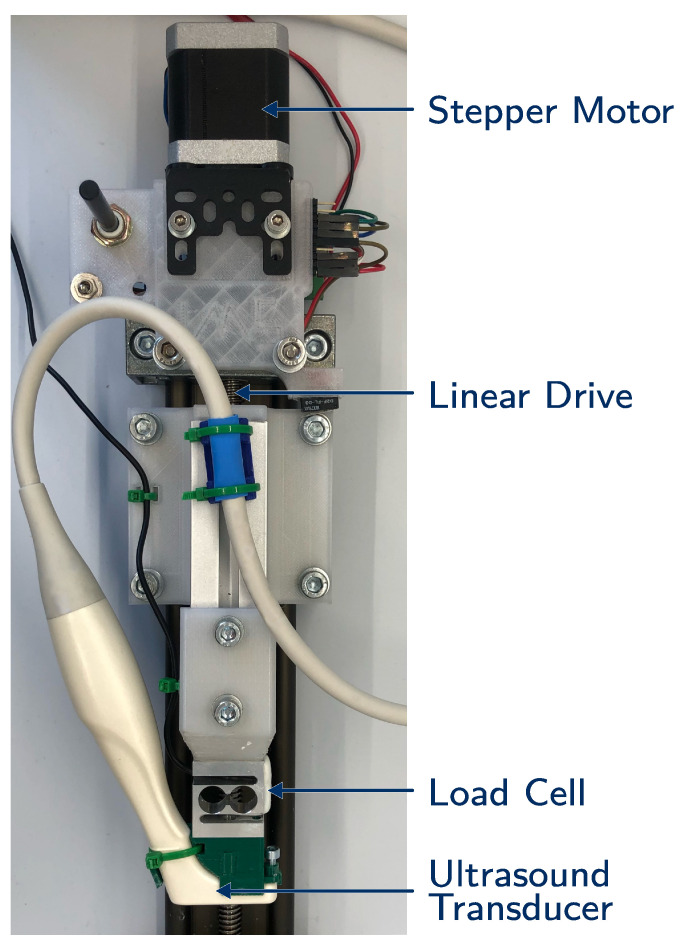
The experimental setup: the stepper motor is used to power the linear drive, which compresses the specimen with the ultrasound transducer. The compressing force is measured by the load cell.

**Figure 2 sensors-21-03010-f002:**
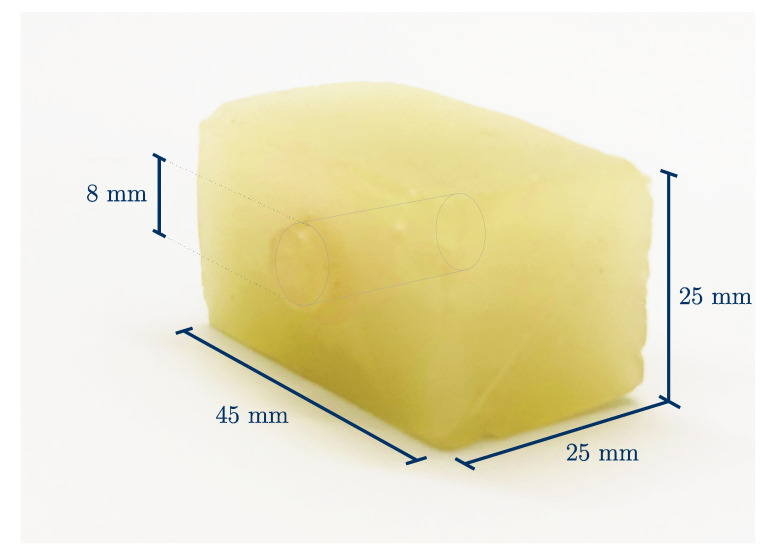
The dimensions of the specimen with inclusion. The homogeneous samples had the same outer dimensions, except that the inclusion was omitted.

**Figure 3 sensors-21-03010-f003:**
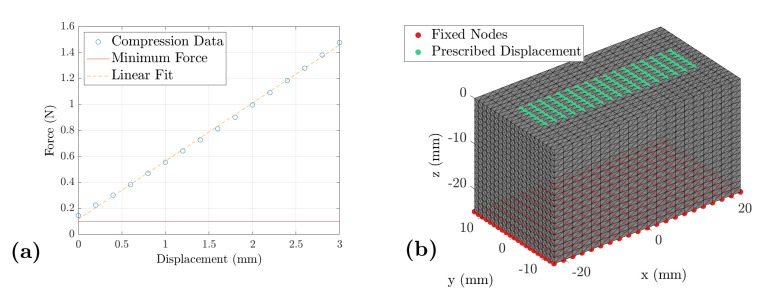
Boundary conditions for the finite element optimization: (**a**) The measured load curve for a gelatin sample: The displacement of the sample surface is plotted against the compression force. Thus, the measured displacement is used as input for the FE model. The defined contact force of 0.1
N and a linear fit are additionally indicated. (**b**) The FE model: The red dots indicate the fixed nodes at the bottom of the gelatin sample and the green dots indicate the moving interface between transducer and gelatin. Furthermore, the green ones show the boundary on which the compressing force is measured and later used as the optimization target.

**Figure 4 sensors-21-03010-f004:**
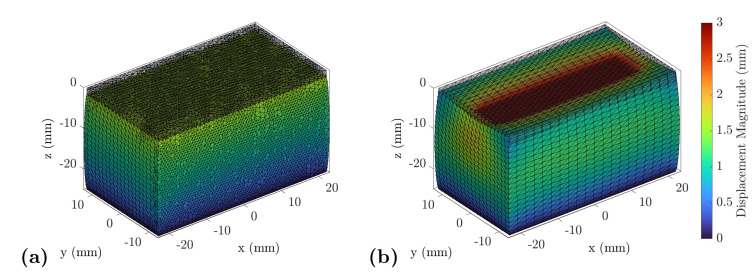
Exemplary result of the forward model: The dimensions of the simulated specimen are plotted on the *x*-, *y*- and *z*-axis, where the coordinate origin was chosen in the center of the compressing object. The color indicates the magnitude of the simulated displacement vector in relation to the initial position. (**a**) The compression of the complete upper surface—in this work, referred to as the compression test. (**b**) The spatially limited compression with the transducer, here called the indentation test.

**Figure 5 sensors-21-03010-f005:**
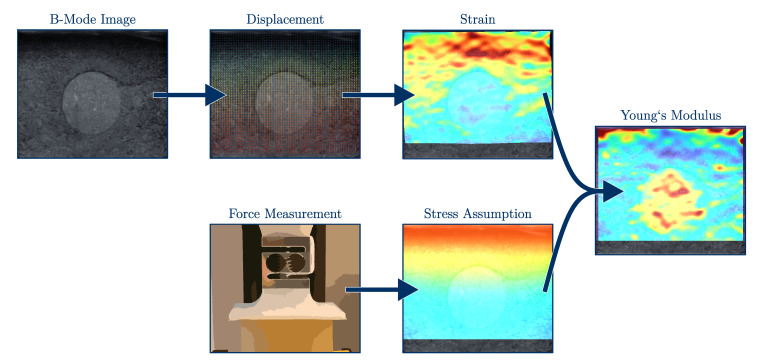
The workflow of the presented elastography algorithm with the basic steps: recording of the B-Mode images, displacement estimation, strain and stress calculation and, conclusively, computation of the Young’s modulus.

**Figure 6 sensors-21-03010-f006:**
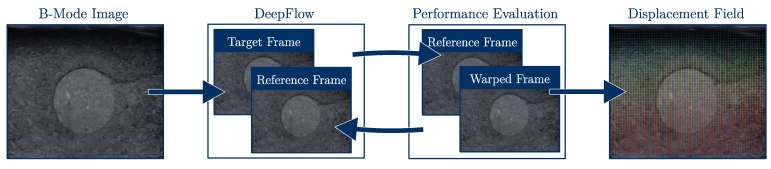
Displacement estimation with automatic redefinition of the reference frame.

**Figure 7 sensors-21-03010-f007:**
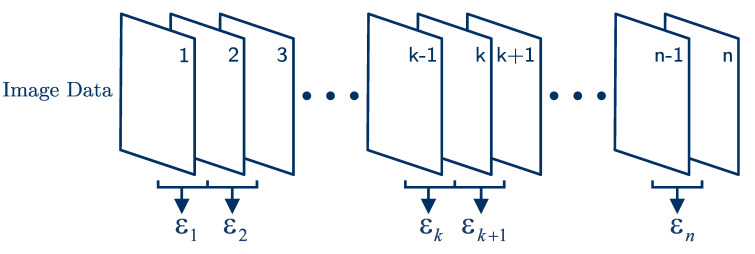
The frame pairing method for strain computation [23].

**Figure 8 sensors-21-03010-f008:**
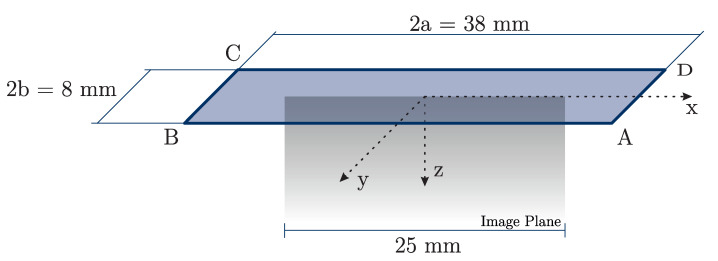
The geometric principal points and the coordinate system for the stress computation. The constant pressure *p* is applied on the area ABCD in positive *z*-direction. In this figure, the contact area of the US transducer with the appropriate dimensions ( 38 mm × 8 mm) is shown. Therefore, the compressing surface is smaller than the top area of the specimen ( 45 mm × 25 mm). Additionally, the image plane is indicated.

**Figure 9 sensors-21-03010-f009:**
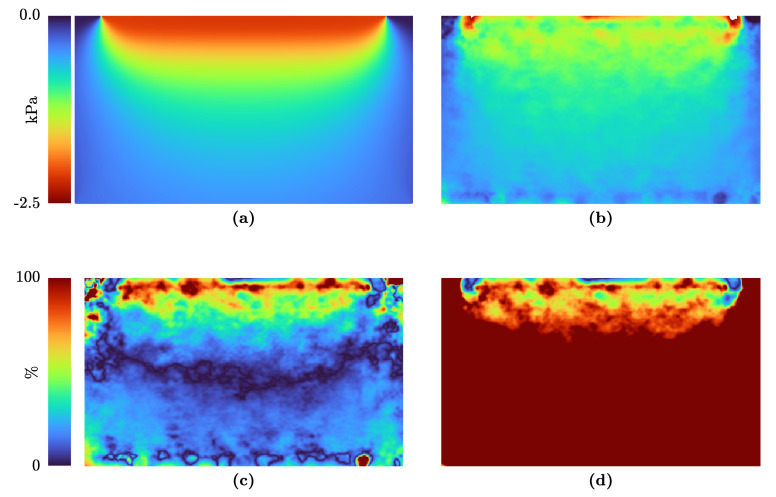
Stress field in a homogeneous specimen under compression of 635 Pa with a transducer width and depth of 38 mm and 8 mm, respectively. The stress in the image plane, which is shown in Figure 8, is displayed: (**a**) Stress field $\sigma_{zz}$ calculated with Love’s solution; (**b**) Stress field $\sigma_{zz}$ estimated by FEBio, further considered to be the true stress distribution; (**c**) Relative error of the stress field estimated by Love’s solution, compared to the results of FEBio; (**d**) Relative error of the stress field with the plane–stress assumption, compared to the results of FEBio.

**Figure 10 sensors-21-03010-f010:**
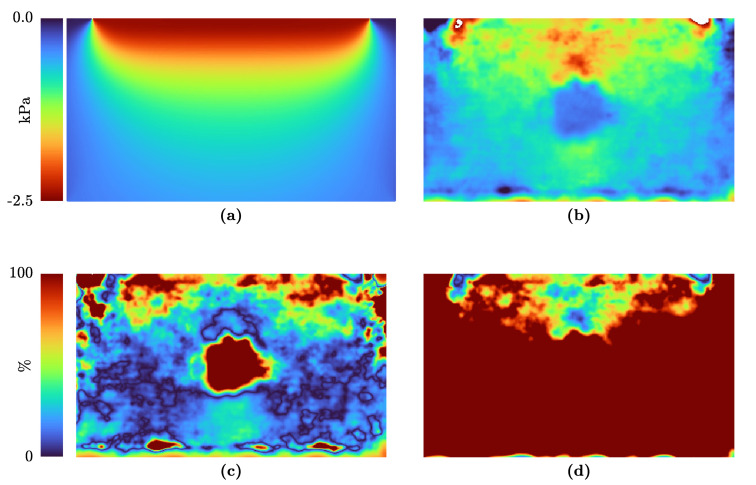
Stress field in a specimen with a stiffer inclusion under compression of 1 kPa with the transducer width and depth of 38 mm and 8 mm, respectively: (**a**) Stress Love’s solution; (**b**) Results of the FE model solved by FEBio; (**c**) Relative error between the FE results and Love’s solution; (**d**) Relative error between the FE results and the plane–stress estimation.

**Figure 11 sensors-21-03010-f011:**
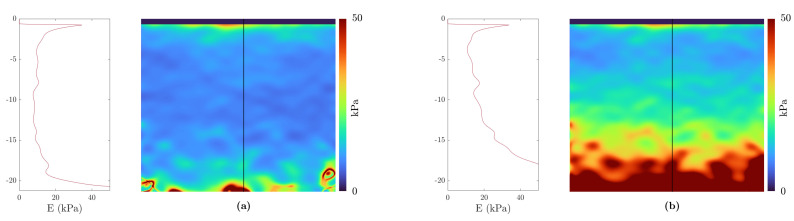
Comparison between the two different stress assumptions of the estimated Young’s modulus of a homogeneous specimen: (**a**) Love’s stress estimation; (**b**) Plane–stress assumption. The black line in the elastogramm marks the cross-section, which is plotted in the left panels.

**Figure 12 sensors-21-03010-f012:**
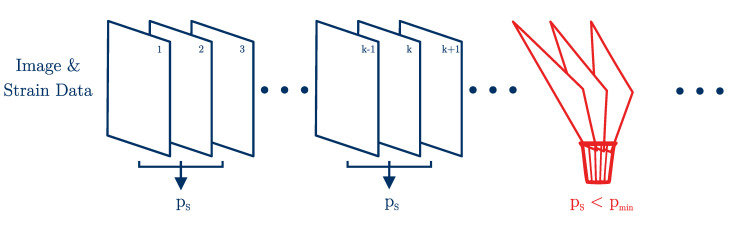
The performance evaluation of image and strain data. Frame pairings which do not fulfill the minimum performance requirement are not considered for the computation of the elastic modulus.

**Figure 13 sensors-21-03010-f013:**
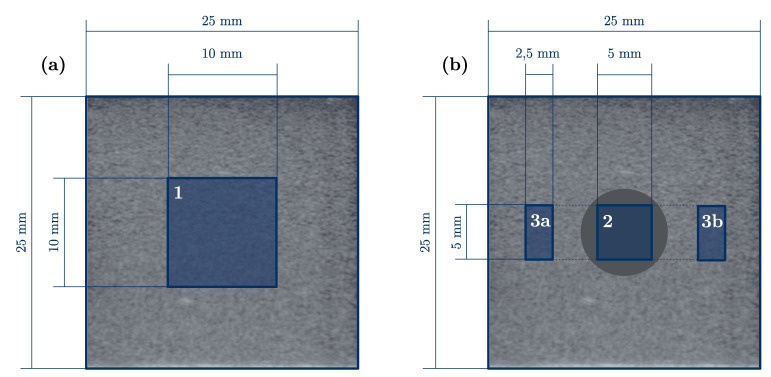
The chosen regions of interest (ROIs) to evaluate the mean Young’s modulus: (**a**) Region 1 is used to evaluate the mean Young’s modulus for homogeneous specimens. (**b**) Region 2 defines the ROI of the inclusion material and the regions 3a,b are combined to calculate the mean Young’s modulus of the body material.

**Figure 14 sensors-21-03010-f014:**
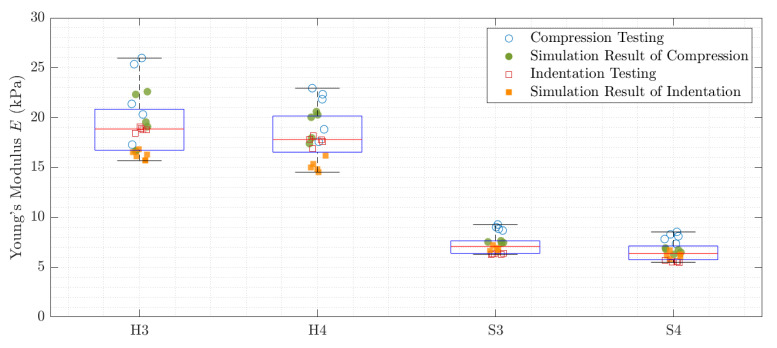
The mechanical evaluation methods are compared to each other in a statistical manner. The red line marks the median of all measurement results and the borders of the box indicate the 25th and 75th percentiles.

**Figure 15 sensors-21-03010-f015:**
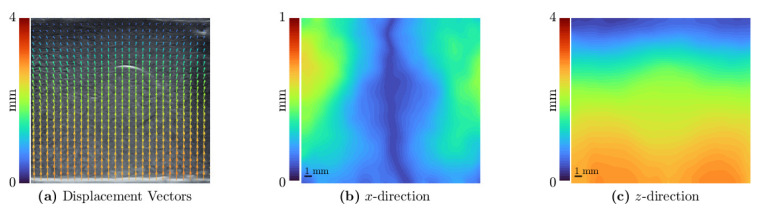
The vectorial displacement in the specimen is shown as an overlay on the ultrasound image in panel (**a**). The absolute value of the displacement in *x*- and *z*-direction is displayed in panel (**b**,**c**), respectively. An indicator of 1 mm is added in the lower left corner. The whole displayed region measures 25 mm × 25 mm.

**Figure 16 sensors-21-03010-f016:**
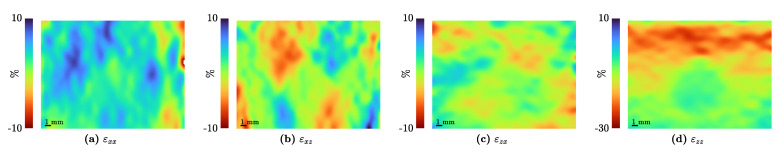
The four strain components in the specimen. The displayed region, compared to Figure 15, is cut, due to the non-valid entries of the convolution of the strain filter. Furthermore, the strain is warped to the actual position of the specimen element at that compression rate.

**Figure 17 sensors-21-03010-f017:**
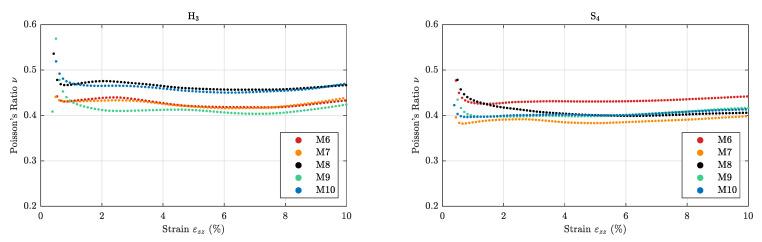
Strain-dependent Poisson’s ratio ν of the homogeneous specimens.

**Figure 18 sensors-21-03010-f018:**
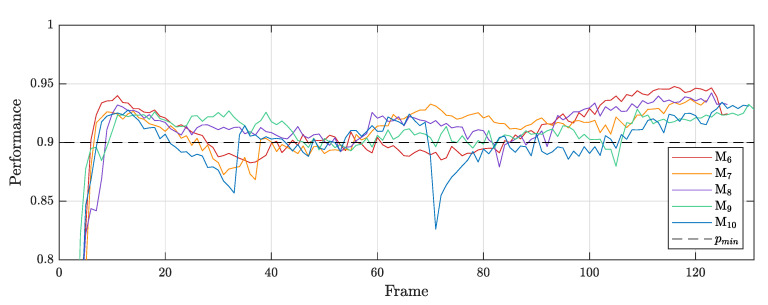
Performance $p_{E}$ of the elastography algorithm of I${}_{4}$.

**Figure 19 sensors-21-03010-f019:**
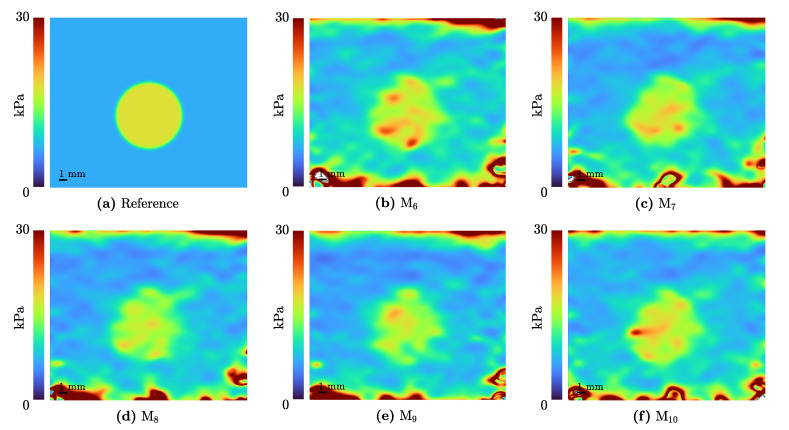
The results of the elastography algorithm of I${}_{4}$ compared to the reference elastogram, which represents the mechanically measured Young’s modulus for the two mixtures.

**Figure 20 sensors-21-03010-f020:**
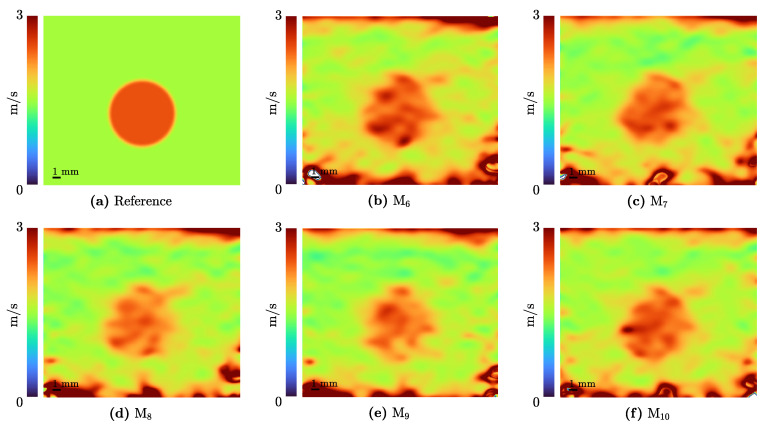
The results of the elastography algorithm of I${}_{4}$ converted to the clinically used shear wave velocity $c_{t}$.

**Figure 21 sensors-21-03010-f021:**
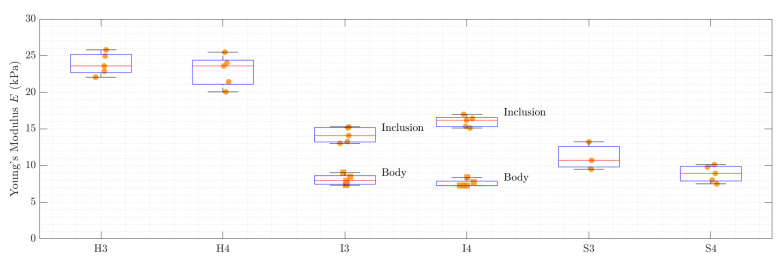
The elastography results for the six specimens. The values indicated here were calculated according to the regions of interest.

**Figure 22 sensors-21-03010-f022:**
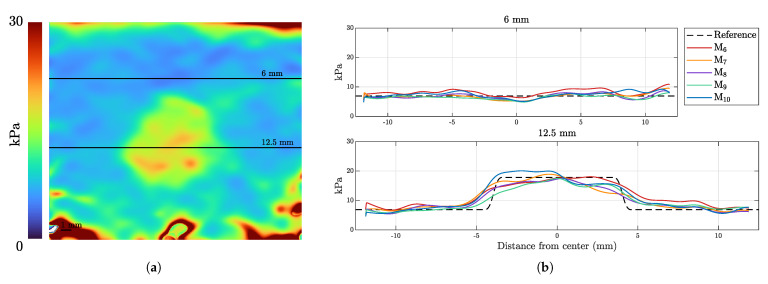
Comparison of the expected Young’s modulus at two defined depths in specimen I${}_{4}$. (**a**) The shown observation depths are measured from the transducer–tissue interface; (**b**) The elastography measurement data are displayed as solid lines and the mechanically measured reference as a dashed line.

**Table 1 sensors-21-03010-t001:** The ingredients of the gelatin mixtures given in percentage by mass.

Mixture	Sample Names	Water %	Gelatin %	Glass Beads %
Soft	$S_{1\ldots n}$	92.5	5	2.5
Hard	$H_{1\ldots n}$	87.5	10	2.5

**Table 2 sensors-21-03010-t002:** Combined results of all mechanical methods and the statistical indicators.

Specimen	Median KPa	Standard Deviation KPa	Maximum Relative Difference %
H${}_{3}$	18.9	2.95	18.7
H${}_{4}$	17.8	2.49	20.2
S${}_{3}$	6.92	1.15	33.9
S${}_{4}$	6.38	0.97	17.0

**Table 3 sensors-21-03010-t003:** Combined results of the elastography algorithm and the statistical indicators.

Specimen	Median KPa	Standard Deviation KPa	Maximum Relative Difference %
H${}_{3}$	23.6	1.52	7.53
H${}_{4}$	23.6	2.16	12.5
I${}_{3}$ (Inclusion)	14.1	1.03	7.92
I${}_{4}$ (Inclusion)	16.2	0.78	5.56
S${}_{3}$	10.7	1.91	14.8
S${}_{4}$	8.94	1.12	15.5
I${}_{3}$ (Body)	7.93	0.71	9.17
I${}_{4}$ (Body)	7.28	0.49	4.29

## Data Availability

Data available on request from the authors.

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
