# Peer review of "A Quasi-Static Quantitative Ultrasound Elastography Algorithm Using Optical Flow"

_sensors, 2021, doi:10.3390/s21093010_

Round 1
Reviewer 1 Report
Relative error of around 34% is quite significant. I see that you mention drying of the phantom as a possible reason but could this be due to viscoelastic nature of the phantom materials that was not considered in this work. I think discussing viscoelastic properties could be an interesting paragraph in the discussion and a good future direction for the work.
Reviewer 2 Report
Strong study! And well structured. I have some suggestions which might help the authors to improve the paper.
Lines 43-47. The structure of the introduction section is a bit disorganized. I would suggest removing the sentences between 43 and 47, and adding to the last paragraph of the introduction section .
Figure4. What does the point 0 represent? Displacement starting point? Please clarify what do the axis x-y-z represent. Displacement magnitude? Figure9. Same question. Where does the displacement start? Generally, clinical guidelines suggest m/s unit. In order to generalize your results to the clinical settings, would it be possible to provide the results in m/s unit? Thank you for considering my recommendations.
Round 2
Reviewer 1 Report
The authors have addressed my comments.
Reviewer 2 Report
Thank you for addressing the comments. The manuscript seems improved. I do not have any further suggestions.